

# Experimental evaluation of the motion-induced effects on turbulent fluctuations measurement on floating lidar systems

Nicolas Thebault [1,*], Maxime Thiébaut [1,*], Marc Le Boulluec [2], Guillaume Damblans [1], Christophe Maisondieu [2], Cristina Benzo [3], and Florent Guinot [1]

[*]These authors contributed equally to this work.
[1]France Énergies Marines, Technopôle Brest-Iroise, 525 Avenue Alexis de Rochon, 29280 Plouzané, France
[2]IFREMER, 1625 Route de Sainte-Anne, 29280 Plouzané, France
[3]Vaisala France SAS, 6A, rue René Razel, Tech Park, CS 70001, 91400 Saclay Cedex, France

**Correspondence:** Maxime Thiébaut (maxime.thiebaut@france-energies-marines.org)

**Abstract.** This study investigates the impact of motion on the line-of-sight (LOS) turbulent velocity fluctuations derived from lidar profiler measurements. Onshore tests were conducted using a WindCube v2.1 lidar, referred to as the mobile lidar, mounted on a hexapod to simulate buoy motion, with a fixed lidar used as a reference. To assess the motion-induced effects on turbulent velocity fluctuations measured by floating lidar systems, the root-mean-square error (RMSE) of LOS velocity fluctuations obtained from the fixed and mobile lidars was calculated. A comprehensive wind dataset spanning 45 hours was analyzed, with a focus on regular motions involving single-axis rotations and combinations of rotations around multiple axes. The investigation of single-axis rotations revealed that the primary influencing factor on the results was the alignment between wind direction and the axis of rotation. The highest RMSE values occurred when winds propagated perpendicular to the rotation axis, resulting in pitch motion, whereas the lowest RMSE values were observed when wind propagated along the rotation axis, leading to roll motion. Furthermore, yaw motion was found to increase the RMSE compared to scenarios without yaw motion. Moreover, the addition of motion around extra axes of rotation was found to increase RMSE. High wind speed emerged as a significant driver of RMSE, with higher velocities leading to higher RMSE values. The study also indicated that the role of wind shear in influencing RMSE of LOS velocity fluctuations requires further investigation. Additionally, the study explored the impact of motion period, revealing that motion frequencies affect the LOS velocity spectra within the expected inertial sub-range. However, the impact on RMSE was found to be limited in comparison to the amplitude, wind direction, and wind speed.

## 1 Introduction

In recent years, wind lidar profilers have emerged as the standard measurement technology, replacing traditional meteorological masts equipped with *in-situ* sensors like anemometers. However, with the expansion of offshore wind projects into deeper water zones, acquiring data from meteorological masts has become prohibitively expensive. To address this challenge, the industry has developed floating lidar systems (FLS), which integrate lidar units onto standalone floating structures, such as buoys. FLS





present an opportunity for cost reduction compared to fixed masts while offering comparable data quality and the ability to measure at the same or even greater heights above the water surface.

FLS have demonstrated their accuracy in measuring wind speed and direction (Smith et al., 2006; Emeis et al., 2007;
Sjöholm et al., 2008; Wagner et al., 2011; Gottschall et al., 2012; Kim et al., 2016)). However, they have not been widely accepted for turbulence measurement. Metrics such as turbulence intensity (TI) are crucial for establishing characteristic design conditions for wind turbines and play a vital role in developing robust design tools that enhance the survivability, reliability, and performance of these machines. Incorporating turbulence measurement capabilities into FLS is an essential step toward realizing these benefits.

Turbulence measurements obtained from lidar profilers are subject to two main systematic errors: underestimation caused by volume averaging (both inter-beam and intra-beam) and overestimation resulting from the cross-contamination effect (Sathe et al., 2011; Sathe and Mann, 2013; Kelberlau and Mann, 2020). These sources of error are independent of each other and, in most cases, their effects do not offset one another. However, there are instances where they do align, resulting in lidar-derived turbulence measurements that align with those obtained from a reference instrument, such as a sonic anemometer. As
a consequence, correct results can be obtained, albeit for the wrong underlying reasons.

Another source of error arises when utilizing FLS. Buoys are subject to translational (surge, sway, and heave) and rotational motions (pitch, roll, and yaw), which can have a detrimental impact on the lidar's measurements. the motions of the float further contribute to introducing high-frequency (in the range of the wave frequencies) fluctuations to the recorded wind data. To address this, the development of motion-compensation algorithms is essential for filtering out these additional fluctuations
in turbulence datasets. Numerous research teams worldwide are actively investigating this research topic (Gutiérrez-Antuñano et al., 2018; Kelberlau et al., 2020; Désert et al., 2021; Salcedo-Bosch et al., 2021, 2022). However, it is important to note that the present paper does not focus on presenting a complete motion-compensation algorithm. Instead, it provides an initial step, which involves assessing the effects of motion on turbulent fluctuation measurements.

In the wind power industry, turbulence intensity (TI) is commonly assessed by calculating the variance of the three velocity
components derived from reconstructed velocity components, which are computed based on the line-of-sight (LOS) velocities. However, for a lidar profiler like the Vaisala WindCube v2.1, an alternative approach is available, which involves computing the variances of the three velocity components based on the variances of the LOS velocity measurements obtained from the five independent beams (Eberhard et al., 1989). The advantages and disadvantages of each methodology are discussed in Peña and Mann (2019).

The future development of the motion-compensation algorithm, based on the measurements presented in this paper, will rely on calculating the variance of the LOS velocity measurements. Therefore, the scope of the present paper is limited to addressing the impact of motion on LOS velocity fluctuations, while the calculation of turbulence intensity based on the variances of the LOS velocity components will serve as a foundation for the forthcoming motion-compensation algorithm.

In the past decade, several studies have examined the impact of motion on wind speed, wind direction, and turbulence
intensity (TI) measurements. For instance, Gottschall et al. (2014) conducted numerical and experimental investigations to analyze the effects of motions originating from various offshore platforms on WindCube v2 lidar measurements. While the



mean wind speed and direction measurements showed little difference compared to fixed reference measurements, TI measured by the mobile lidar was consistently higher. These findings were later supported by Gutiérrez-Antuñano et al. (2018), who utilized a software-based motion simulator to replicate the velocity azimuth display of a ZephIR 300 lidar. They employed a

moment-computation recursive procedure to estimate the motion-induced error standard deviation in horizontal wind speed, as well as the motion-induced TI, under simple-harmonic motion conditions.

Additionally, Kelberlau and Mann (2022) utilized numerical and analytical methods to quantitatively assess the bias in lidar wind speed measurements. Their findings revealed that the average bias was dependent on several factors, including the amplitude and frequency of motion, the angle between motion and wind direction, as well as wind speed and the strength of

wind shear. Moreover, a recent study by Gräfe et al. (2023) identified pitch as the primary motion factor influencing wind speed measurements obtained through nacelle-based lidar.

These studies collectively emphasize the significance of accounting for motion-induced effects on wind measurements and offer valuable insights into the specific factors that impact the accuracy and reliability of lidar-based measurements in various motion scenarios. Their findings contribute to enhancing our understanding of the complexities involved in mitigating motion-

related errors and improving the precision of lidar-based wind measurements.

An alternative method to investigate the impact of motion in an experiment involves placing a wind lidar on a moving platform and comparing the measurements with data obtained from a stationary lidar system of the same type positioned nearby. Hellevang and Reuder (2013) present their findings on two different lidar models (WindCube and ZephIR) and various motion scenarios. Unfortunately, the chosen motion patterns do not align with typical FLS, and the duration of each test case

is too short to accurately quantify the measurement error induced by motion. Tiana-Alsina et al. (2015) also utilize a ZephIR lidar in different scenarios, but only for brief time periods, making it challenging to statistically assess the minor error caused by motion.

Further investigations are necessary to conduct experimental testing onshore using motion platforms that simulate typical sea motions. Such experiments would provide confirmation of the findings obtained from simulation tools employed in previous

studies. In this study, our objective is to address this research gap by quantifying wind fluctuations measured by a lidar mounted on a moving platform and comparing them to a reference fixed lidar. To recreate potential buoy movements, we employed an hexapod within a controlled environment. Initially, we focused on rotations around a single axis, varying the amplitudes and periods. Subsequently, we explored the impact of coupling two and three rotational degrees of freedom on LOS velocity fluctuations measured by the mobile lidar. The experimental campaign was conducted onshore over several weeks to gather

measurements encompassing different wind speed ranges.

## 2 Data and method

### 2.1 Experimental campaign

The experimental campaign was conducted at Ifremer's site in Brest, France. The experimental setup consisted of two Wind-Cube v2.1 lidars deployed onshore. One lidar was installed in a fixed configuration, providing reference measurements, while



| Cycle | Date (2022) | Starting hour (UTC 0) | Wind speed (m/s) | Wind direction (°) |
|-------|-------------|-----------------------|------------------|--------------------|
| C1 | Oct., 6th | 7:39 | $3 \pm 2$ | $208 \pm 19$ |
| C2 | Oct., 6th | 11:17 | $6 \pm 1$ | $225 \pm 5$ |
| C3 | Oct., 11th | 7:22 | $7 \pm 1$ | $72 \pm 8$ |
| C4 | Oct., 11th | 10:30 | $4 \pm 1$ | $77 \pm 20$ |
| C5 | Oct., 12th | 7:28 | $4 \pm 1$ | $215 \pm 20$ |
| C6 | Oct., 12th | 10:34 | $6 \pm 1$ | $217 \pm 8$ |
| C7 | Oct., 17th | 8:35 | $3 \pm 1$ | $208 \pm 14$ |
| C8 | Oct., 18th | 10:32 | $10 \pm 1$ | $106 \pm 4$ |
| C9 | Oct., 19th | 7:36 | $10 \pm 1$ | $110 \pm 10$ |
| C10 | Oct., 19th | 10:50 | $11 \pm 2$ | $146 \pm 15$ |
| C11 | Oct., 26th | 8:04 | $14 \pm 1$ | $203 \pm 5$ |
| C12 | Oct., 26th | 10:40 | $14 \pm 1$ | $200 \pm 6$ |
| C13 | Oct., 28th | 7:34 | $7 \pm 1$ | $227 \pm 6$ |
| C14 | Nov., 9th | 9:03 | $9 \pm 1$ | $252 \pm 8$ |
| C15 | Nov., 9th | 12:20 | $9 \pm 1$ | $251 \pm 9$ |

**Table 1.** Date and starting hour for each cycle. The cycles are 3 hours long. The mean wind speed and direction at 140 m elevation is given with the standard deviation.

the second lidar was installed on a Stewart platform, also known as a hexapod. The lidar installed on the hexapod is referred to as the "mobile lidar." The hexapod used in the experiment is the Mistral 800 by SYMETRIE. It consists of a lower (base) platform and an upper (nacelle) platform that can perform movements along all six degrees of freedom (Fig. 1a). The maximum range of motion and dynamic capabilities of the hexapod are detailed in the appendix (Table A1). The hexapod enables a wide range of courses, speeds, and accelerations for rotational motions, although the translation motions with the hexapod are

limited due to the shorter courses enabled by the technology. Specifically, rotations around the x, y, and z axes are referred to as Rx, Ry, and Rz, respectively.

The two lidars were positioned approximately 10 meters apart from each other. To ensure their protection, the mobile lidar and the hexapod were housed inside a container (Fig. 1). It is important to note that the hexapod and its connector are sensitive to humidity and therefore cannot be operated during rainy conditions. Beams 1 and 3 of both lidars were aligned with the $y$-axis

of the hexapod, while beams 2 and 4 were aligned with the $x$-axis (Fig. 1b). Beam 5 was directed vertically upward along the $z$-axis. The LOS velocity data of a standard commercial WindCube v2.1 lidar has a sampling rate of 0.25 Hz. However, in this experiment, a prototype configuration of the lidar with a four times faster sampling rate of 1 Hz was used. The inertial unit located on the hexapod had a sampling rate of 100 Hz.

For power supply, the lidars were connected to electricity in a building situated approximately 15 meters southeast of the

working area. Both lidars were installed at a height of 3 meters above the ground (Fig. 1).

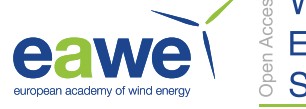


**Figure 1.** (a) General view of the hexapod along with its main components. The (xyz) coordinate system, along with the associated rotational motions, utilized in this study, are also illustrated. (b) Top view of the hexapod showcases the positioning of the WindCube v2.1 mobile lidar. (c) Overall view of the experimental campaign. (d) The two lidars at IFREMER test site. Foreground, the mobile lidar mounted on the hexapod installed in the container and, background, the fixed lidar.





| Sequence | Ry | Rx | Rz | Corresponding Scenario |
|---|---|---|---|---|
| S1 | T = 4 s, A = 5° | NA | NA | Floater at high natural period, weak response below |
| S2 | T = 4 s, A = 15° | NA | NA | Typical large Buoy natural period, strong response |
| S3 | T = 6 s, A = 5° | NA | NA | Small buoy natural period, weak response above |
| S4 | T = 6 s, A = 15° | NA | NA | Large or spar buoy |
| S5 | T = 8 s, A = 5° | NA | NA | Small buoy natural period, weak response above |
| S6 | T = 8 s, A = 15° | NA | NA | Very Large or deep spar buoy |
| S7 | T = 6 s, A = 5° | T = 6 s, A = 5° | NA | Small buoy /quarter seas / 3 semi-taut lines |
| S8 | T = 6 s, A = 5° | NA | T = 6 s, A = 5° | Small buoy /quarter seas / 3 lines |
| S9 | T = 6 s, A = 5° | T = 6 s A = 5° | T = 6 s, A = 5° | Small buoy /quarter seas / 3 lines |

**Table 2.** Hexapod Motion definition (T = Period, A = Amplitude, NA = Not Applicable): The first 6 sequences consist of rotations Ry around the y-axis with varying amplitudes and periods. The following 3 sequences involve coupled rotations around the x, y, and z-axes. Each sequence is associated with a specific scenario corresponding to different types of floaters.

The fixed lidar operated continuously for a duration of 57 days, spanning from September 14th to November 9th, 2022. On the other hand, the mobile lidar was activated and deactivated on specific dates within the same period to avoid data collection during rainy days (Table 1, Fig. 4a). For this study, our focus lies on the 15 measurement cycles recorded between October 6th, 2022, and November 9th, 2022. Each cycle consisted of 12 sequences, with nine 10-minute "regular" sequences involving rotations around the y-axis at various amplitudes and periods, as well as movements that coupled rotations around different axes, namely (i) x and y, (ii) y and z, and (iii) x, y, and z (Table 2). Following these regular sequences, there were three 30-minute sequences, two consisting in random motions induced by real sea-states and one representing "pink noise." However, these latter sequences were not used for the purposes of this study and will be utilized later for validating the motion-compensation algorithm.

The signals from both lidars were divided into 10-minute ensembles, each containing 600 data points. The two lidars recorded data at 10 different elevations, evenly distributed between 40 m and 220 m. Throughout the measurement campaign, the data availability remained at 100%.

At the test site, wind characteristics captured by the fixed lidar over the 57-day deployment period indicated wind speeds ranging between 0.5 m/s and 23.3 m/s, with a mean of 10 m/s at 140 m elevation. The prevailing wind directions were predominantly from two main sectors: the South-West and the South-East (indicated by gray shading in Fig. 3). These sectors align with the average wind directions recorded during the 15 measurement cycles of the mobile lidar (depicted by boxes in Fig. 3). Moreover, the 15 cycles encompassed various wind speed ranges, with speeds ranging from 1.2 m/s to 17 m/s and a mean of 7.5 m/s at 140 m elevation (Table 1, Fig. 4b).

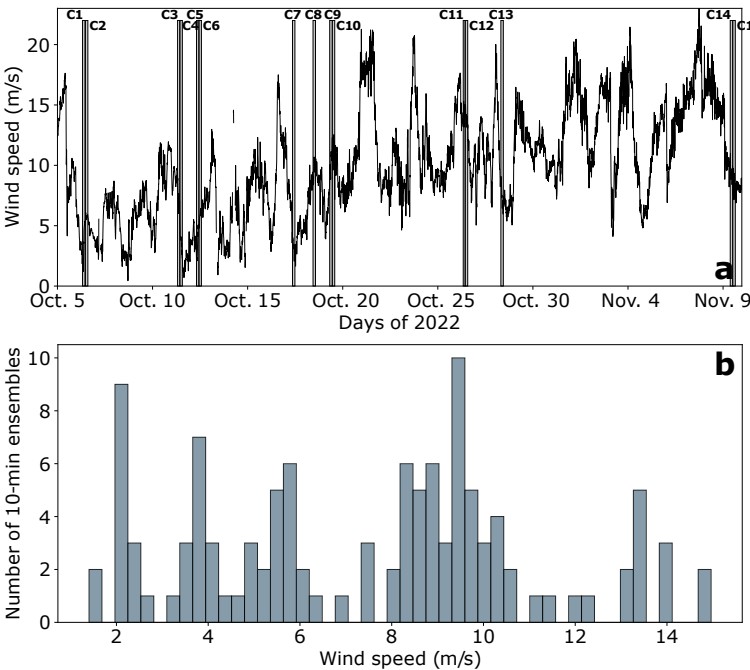

**Figure 2.** (a) 10-Minute averaged wind speed measured by the fixed lidar. The gray areas indicate the time periods of data acquisition by the mobile lidar. (b) Wind speed distribution recorded by the fixed lidar during the data acquisition periods of the mobile lidar. Wind statistics presented in each panel were obtained at an elevation of 140 m.

## 2.2 Motion specifications

The hexapod was configured to simulate the possible motions of a moored floater. France Energies Marines developed a large buoy, known as MONABIOP, deployed at the Mistral test site in the Mediterranean Sea. To replicate the quasi-static loading of turbine thrust, the buoy is moored using three semi-taut lines, which restrict its motion. Therefore, it can be assumed that this buoy technology accurately represents the motion dynamics of a floating lidar system. The MONABIOP buoy is specifically designed for testing and validating a mooring system that utilizes nylon ropes. The buoy's dynamics, mainly governed by the

tilt having a natural period of 4 seconds was used as a reference for defining different motion scenarios for the hexapod. These scenarios aimed to represent standard floater motions, as well as larger or specific technologies such as SPAR buoys.

The amplitude of tilt motion experienced by a FLS is dependent on the prevailing sea state. In very calm seas, minimal dynamic tilt motion is anticipated. However, in the presence of strong waves, the floating platform experiences significant excitation, resulting in larger tilt motions. Amplitudes of 5 deg. and 15 deg. were selected to represent medium and high tilt

motions, respectively. These values accurately reflect the tilt amplitudes typically observed in FLS under specific conditions (Kelberlau and Mann, 2022).



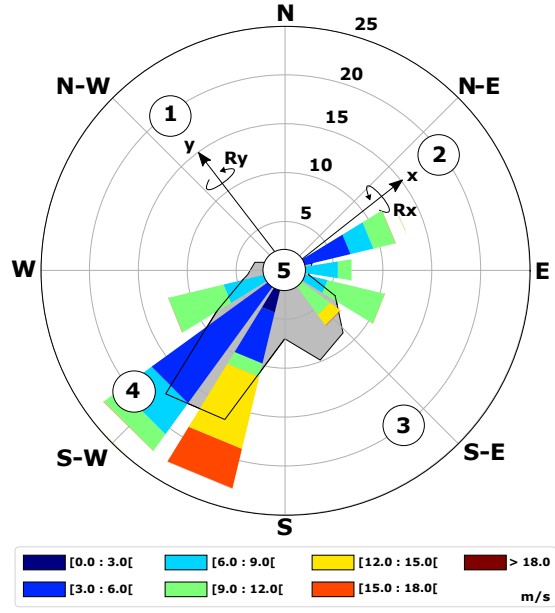

**Figure 3.** Wind rose (color-coded) depicting the results of the 15 measurement cycles conducted using the 9 sequences at 10 different heights. The gray shading with black contour represents the 10-minute mean wind direction at 140 m, as recorded by the fixed lidar over the course of the 57-day deployment. The numbers 1 to 5 indicate the orientation of the lidar's five beams. The x and y axes of the hexapod, corresponding to rotations Rx and Ry, respectively, are also displayed.

The tilt motion period of a FLS is type specific and determined by its mass and hydrodynamic properties. Three periods were chosen: the natural period of the MONABIOP buoy (4 s), twice its natural period (8 s), and an intermediate value (6 s).

Consequently, nine cases were chosen to cover a wide range of motion scenarios while minimizing the number of cases needed, ensuring they are representative of various floating lidar technologies (Table 2). Initially, rotations around a single axis were applied, followed by the gradual addition of rotations around one and two other axes.

### 2.3 Assessing motion-induced effects: evaluation metrics

The impact of motion-induced effects on turbulent velocity fluctuations measured by floating lidar systems was assessed by calculating the root-mean-square error (RMSE) of the turbulent velocity fluctuations obtained from the fixed and mobile lidars. The turbulent velocity fluctuations were estimated by calculating the standard deviation, $\sigma$, of the mean-detrended signal for the 10-minute ensembles of the LOS velocities. The focus was on beam 1 and beam 2, which are positioned at 90 degrees to each other (Fig. 3).

The initial measurements from the first 6 sequences, spanning 15 cycles, were specifically chosen to examine the impact of rotation amplitude and period around the y-axis (refer to Table 2), on the RMSE of the LOS velocity fluctuations. Additionally,





|  | S1 | S2 | S3 | S4 | S5 | S6 | S7 | S8 | S9 |
|---|---|---|---|---|---|---|---|---|---|
| Beam 1 | 0.17 | 0.69 | 0.16 | 0.67 | 0.14 | 0.61 | 0.16 | 0.14 | 0.18 |
| Beam 2 | 0.17 | 0.67 | 0.18 | 0.63 | 0.14 | 0.59 | 0.16 | 0.13 | 0.13 |
| Beam 3 | 0.18 | 0.73 | 0.19 | 0.68 | 0.16 | 0.63 | 0.17 | 0.16 | 0.21 |
| Beam 4 | 0.15 | 0.75 | 0.19 | 0.64 | 0.14 | 0.56 | 0.13 | 0.12 | 0.18 |
| Beam 5 | 0.26 | 0.92 | 0.30 | 0.88 | 0.24 | 0.79 | 0.27 | 0.21 | 0.26 |

**Table 3.** RMSE values (m/s) for each beam and sequence (S). The mean RMSE is calculated as an average across all 15 cycles and 10 measurement heights.

the impact of various factors such as wind speed, wind shear, wind direction, was investigated. Following this analysis, the effects of coupling multiple rotations were further evaluated using sequences 7, 8, and 9 of the 15 cycles.

## 3 Results

### 3.1 Preliminary observations

The study begins with a comparison of LOS velocity time series obtained from the fixed and mobile lidars. Fig. 4 illustrates this analysis for the sequence 1 of the cycle 13 at an elevation of 140 m. The comparison, beam by beam, reveals that the time series measured by the fixed and mobile lidars closely align with each other. Notably, the velocity spikes observed in the fixed lidar measurements are accurately captured by the mobile lidar. Moreover, the oscillations observed in the mobile lidar measurements clearly indicate the impact of regular motion on the device. In this particular sequence, these oscillations result in a standard deviation of the LOS velocities that is twice as high as the standard deviation computed from the fixed lidar measurements.

Fig. 5 illustrates that, with few exceptions, there is a consistent overestimation of $\sigma$ measured by the mobile lidar compared to $\sigma$ derived from the fixed lidar. When considering all measurement heights and sequences together, the mean $\sigma$ measured by the fixed lidar was found to be 0.45 m/s, whereas $\sigma$ derived from the mobile lidar measurements was approximately 70% higher.

Table 3 presents the mean RMSE values associated with beams 1 and 2, which are similar to the values of beams 3 and 4. However, the mean RMSE values associated with beam 5 were, on average, 46% higher. Notably, sequences 2, 4, and 6 exhibited the highest RMSE values, nearly three times higher than the RMSE calculated for the other sequences. These three sequences correspond to scenarios with 15 deg. amplitude, i.e., three times higher than the other sequences (Table 2).

### 3.2 Impact of motion amplitude and wind speed

Fig. 6 shows the RMSE of the LOS velocity fluctuations for beams 1 and 2 as a function of wind speed measured at all measurement heights. For both beams, the RMSE distributions are similar. Fig. 6a clearly demonstrates that amplitude has


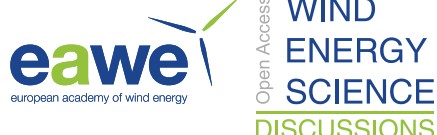

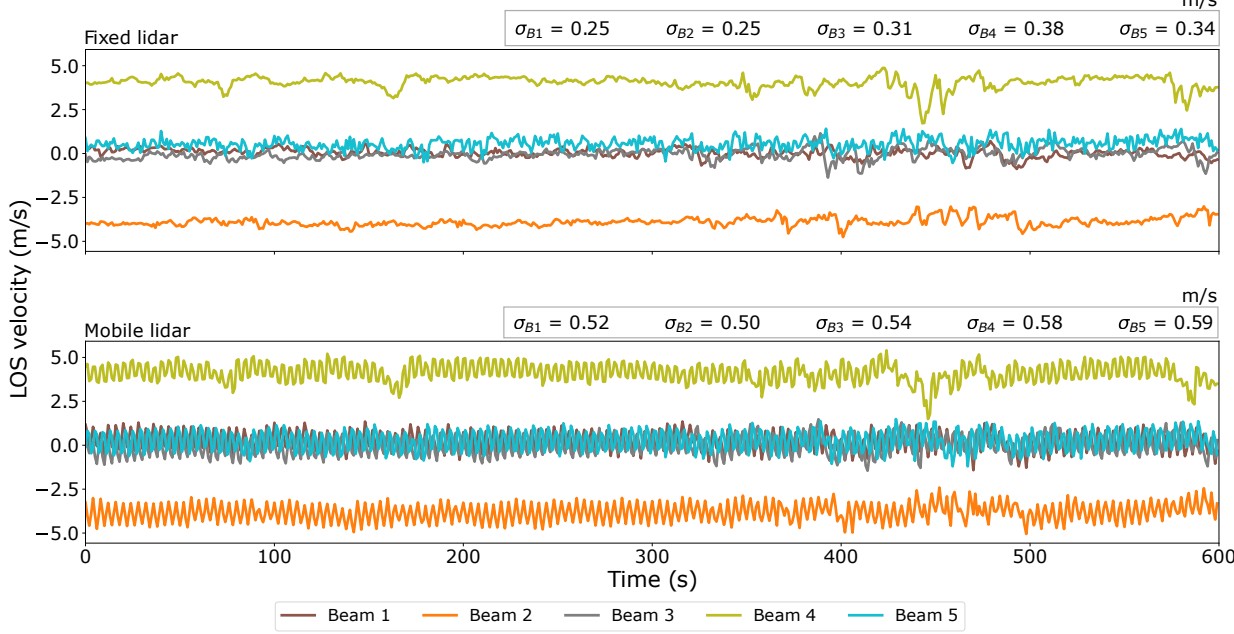

**Figure 4.** LOS velocity time series measured by the 5 beams of the fixed and mobile lidars at an elevation of 140 m during the first sequence of cycle 13. Additionally, the mean standard deviation corresponding to each beam for this specific sequence is also provided.

the primary impact on RMSE. For a given period, higher amplitudes result in higher RMSE values. At a lower amplitude (5 deg.), the mean RMSE associated with beam 1 is 0.15 m/s. However, for an amplitude three times higher, the mean RMSE increases to more than four times higher (0.65 m/s). Additionally, the RMSE exhibits a linear increase with wind speed which

demonstrates that the latter metric is a driver of the magnitude of the RMSE. For the lowest amplitude, the RMSE associated with beam 1 reaches 0.5 m/s at higher wind speeds. In contrast, for the highest amplitude, the RMSE reaches 2 m/s.

### 3.3 Impact of motion period

The period has a lesser impact on wind fluctuation measurements compared to amplitude. The mean RMSE associated with a 4-second period was found to be 0.43 m/s, which is 14% lower than the corresponding value for a period twice as long. The

motion period significantly influences the LOS velocity spectra, as demonstrated in Fig. 8. The mean spectrum is presented, averaged over the 15 cycles and measured at 140 m by beam 1 of both the mobile and fixed lidars. These measurements were taken during sequences 2, 4, and 6, corresponding to motion periods of 4 s, 6 s, and 8 s, respectively, with an amplitude of 15 degrees. The spectra obtained from the mobile lidar clearly exhibit a spike in energy corresponding to the rotation frequency. The height of this spike remains consistent for each motion period. Additionally, it is noteworthy that, at frequencies lower

than that of the spike, the spectral energy derived from measurements of the fixed lidar consistently exceeds that of the mobile lidar. However, once the spike emerges, the opposite occurs, and the mobile lidar's spectral energy surpasses that of the fixed

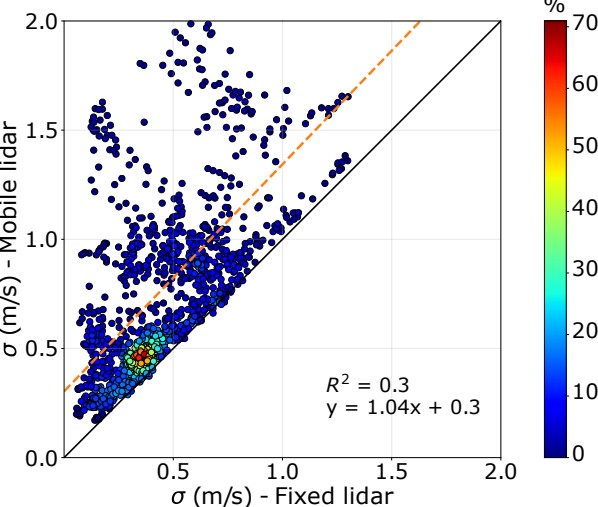

**Figure 5.** Scatter plots, color-coded based on point density, displaying the standard deviation ($\sigma$) of the line-of-sight (LOS) velocity measured by beam 1 at all measurement heights for both the mobile lidar and the fixed lidar. These results are obtained from the 9 sequences used in the 15 measurement cycles. The solid black line represents the line of perfect agreement between the LOS standard deviation derived from the mobile lidar and the fixed lidar. The orange dashed line represents the best fit for the scatter plot.

lidar for the higher frequencies. These findings highlight the influence of motion period on LOS velocity spectra and indicate distinct characteristics in the spectral energy profiles between the mobile and fixed lidar measurements.

### 3.4 Impact of wind direction

The wind roses of RMSE associated with beam 1 and beam 2 (Fig. 7) reveal that the highest RMSE values are observed when the wind is orthogonal ($\pm$ 5°) to the y-axis of rotation. Comparing this polar distribution of RMSE with the polar distribution of wind speed (Fig. 3), we find that although the wind speed is lower when the wind is orthogonal to the y-axis (averaging 6 m/s) compared to when the wind is aligned ($\pm$ 5°) with the y-axis (averaging 7.6 m/s), the mean RMSE is more than 10 times higher for wind orthogonal to the axis of rotation.

### 3.5 Impact of elevation

Fig. 9 illustrates a significant trend: the RMSE shows an upward trend as the elevation increases. Notably, at an altitude of 220 m, the median RMSE exceeds the median value of 0.15 m/s computed at 60 m by more than 2.5 times. This observation aligns with the expected outcome, as previous findings in this study have demonstrated that higher wind speeds lead to higher RMSE in velocity fluctuations.

Although the mean wind speed at 220 m was found to be 30% higher than the wind speed computed at 60 m, it is essential to acknowledge that wind speed alone may not be the sole factor influencing the RMSE of LOS velocity fluctuations when

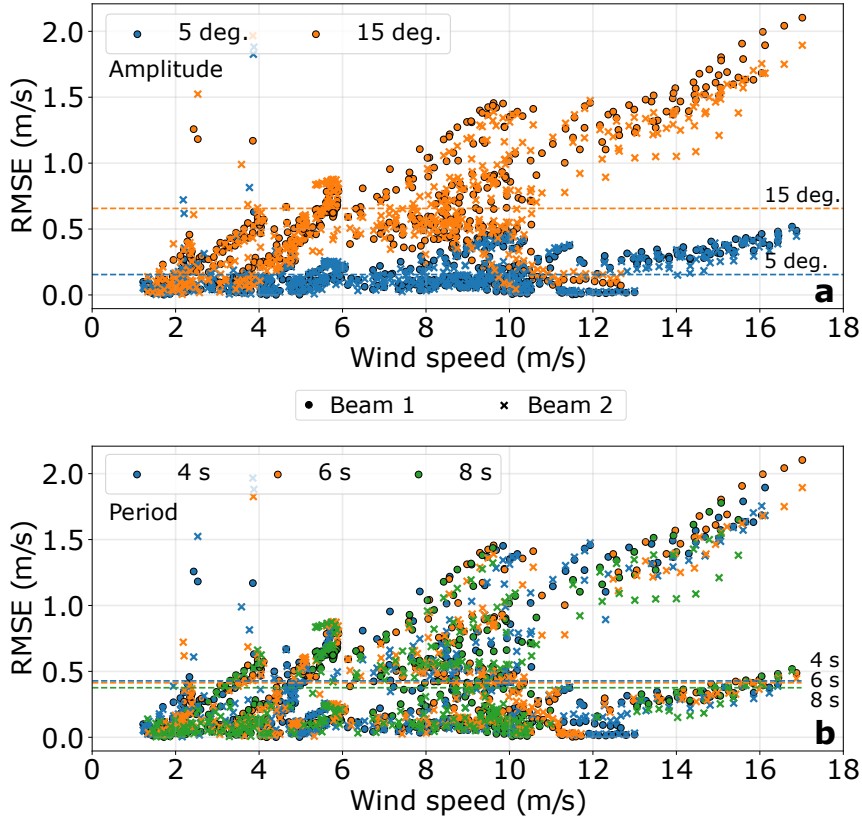

**Figure 6.** Scatter plots depicting the RMSE of LOS velocity fluctuations measured by beam 1 and beam 2 at all measurement heights, plotted against wind speed. These results are obtained from the initial 6 sequences of the 15 measurement cycles, where the motion was specifically designed to replicate rotation around the y-axis with varying amplitudes and periods (refer to Table 2 and Fig. 1). In this context, the term "Amplitude" (and "Period") denotes the investigation of different motion amplitudes (and periods), while setting the motion period (and amplitude) to a single value. In panels (a) and (b), colored horizontal dashed lines illustrate the mean of each distribution solely for beam 1. These lines are accompanied by labels to assist with the interpretation of the figure.

considering the impact of elevation. The rotational displacements of the mobile lidar cause the beam direction to tilt compared to the fixed lidar, resulting in the shifting positions of focus points both vertically and horizontally. Consequently, the mobile lidar does not scan the same volume of air as the fixed lidar, potentially missing out on sampling the same eddies. This effect becomes more pronounced with increasing elevation. Moreover, the increase of the RMSE at higher elevations might also be due to the vertical gradient of the horizontal mean wind speed, i.e., the wind shear which is known to impact the wind vector measured by a FLS (Kelberlau and Mann, 2022).



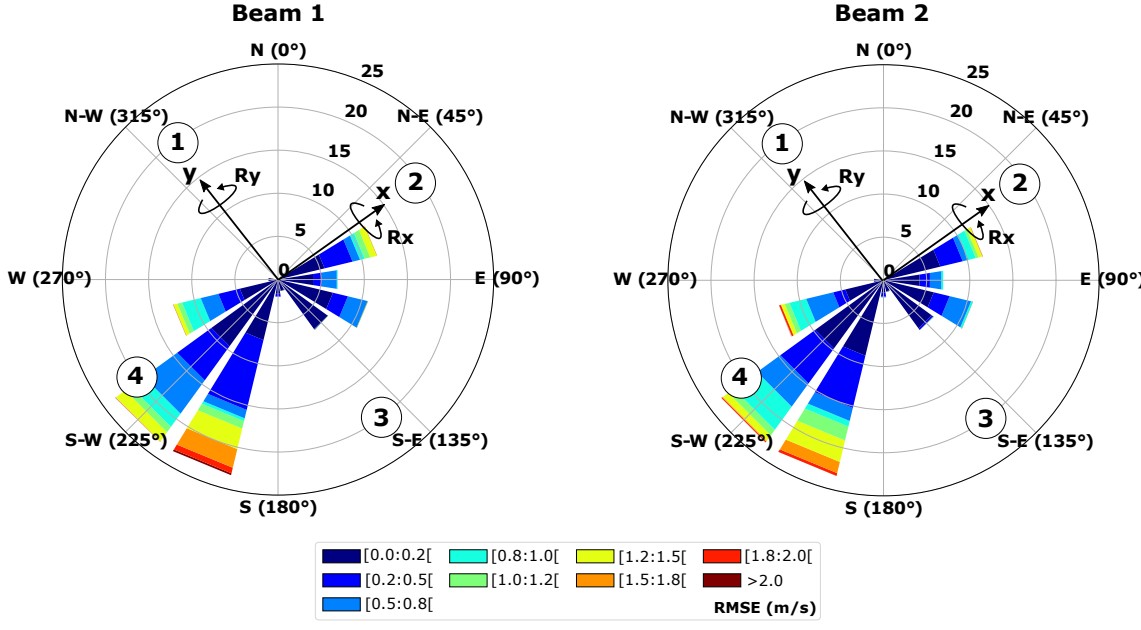

**Figure 7.** Wind rose depicting the RMSE of LOS velocity fluctuations measured by beam 1 and beam 2 at all measurement heights. These results are derived from the initial 6 sequences of the 15 measurement cycles. The 4 numbers displayed correspond to the orientation of the lidar's first four beams. Additionally, the 2 arrows, labeled x and y, represent the orientation of the hexapod.

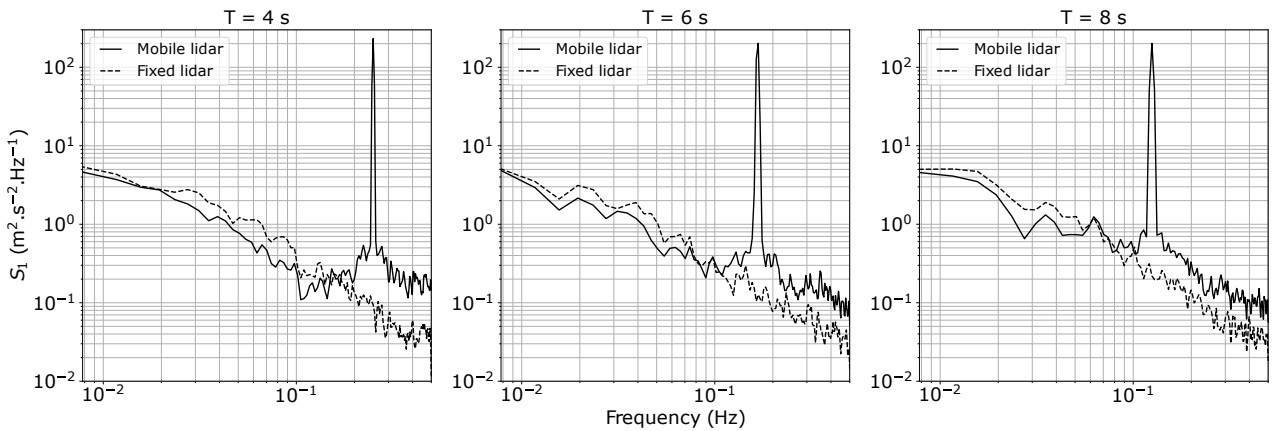

**Figure 8.** Mean spectrum, averaged over the 15 cycles and measured at 140 m by the beam 1 of the mobile (solid line) and fixed (dashed line) lidars during the sequences 2, 4 and 6, corresponding respectively to motion periods of 4 s, 6 s and 8 s with an amplitude of 15 deg.





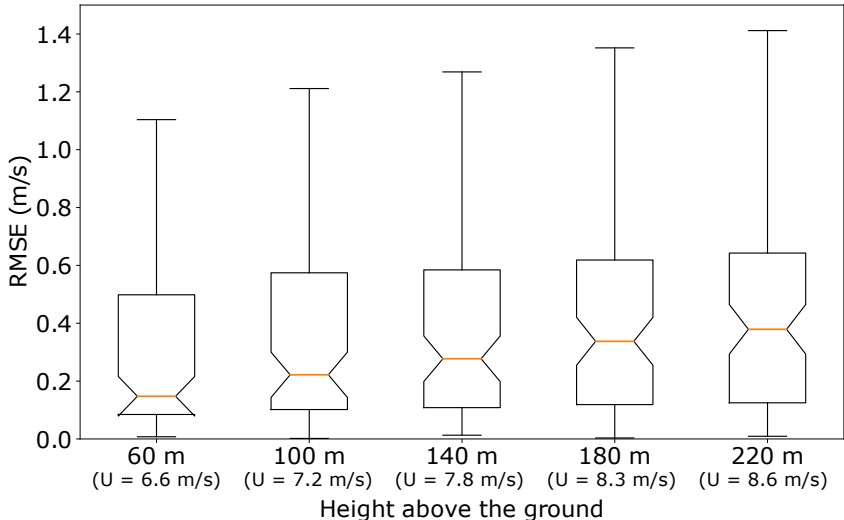

**Figure 9.** Box plot showcasing the RMSE of the LOS velocity fluctuations measured by beam 1 at 5 specific heights. These results are obtained from the initial 6 sequences of the 15 measurement cycles, where the motion was specifically designed to replicate rotation around the y-axis with varying amplitudes and periods (refer to Table 2 and Fig. 1). The mean wind speed, denoted as $U$, is provided for each height.

### 3.6 Impact of wind shear

In the presence of a sheared wind speed profile with usually higher wind speeds at higher elevations, the changes in elevation
have an influence on the mean wind speed. In this study, we assessed the impact of wind shear on the RMSE of the LOS velocity fluctuations by examining the first 6 sequences out of the total 15 cycles.

To determine the wind shear exponent, $\alpha$, we employed individual 10-minute average wind speed vertical profiles derived from measurements obtained by the fixed lidar. These profiles were then fitted using the power law profile recommended by the IEC 61400-3-1 international standard. The fittings yielded a mean relative error of less than 1% (results not shown).

Fig. 10 shows that the height-averaged RMSE of the LOS velocity fluctuations is not governed by the wind shear exponent which varies from -0.05 to 0.55. The RMSE associated with one single rotation of 5 deg. amplitude shows a slight variation around the mean of 0.15 m/s for the entire range of the computed wind shear exponents. Similar results are found for the RMSE associated with one single rotation of 15 deg. amplitude with however a more scattered distribution around the mean of 0.65 m/s and extremes values higher than 1.25 m/s. Those extreme values are found for a wind shear exponent of 0.2 on average
and are associated to wind speed ranging between 13 and 14.5 m/s and wind direction perpendicular to the y-axis of rotation.

### 3.7 Impact of the coupling of motions around several axis of rotation

The wind speeds in the range of 12 to 14 m/s are particularly suitable for comparing the different scenarios involving rotations around one or multiple axes (Fig. 11). Within this range, a general observation is that the coupling involving the three axes of



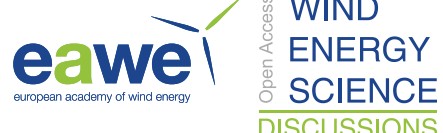

rotation results in the highest RMSE, followed by rotations around the x and y-axes, then rotations around the y and z-axes,
and finally, the single rotation around the y-axis.

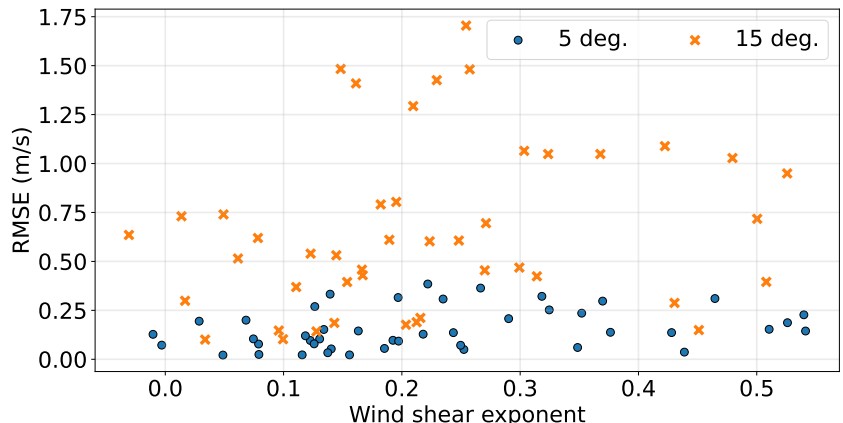

**Figure 10.** Height-averaged RMSE of the LOS velocity fluctuations plotted against the wind shear exponent, based on measurements from the first 6 sequences of the 15 cycles. The data is differentiated for motion associated with 5 deg. (blue dots) and 15 deg. (orange crosses) amplitude.

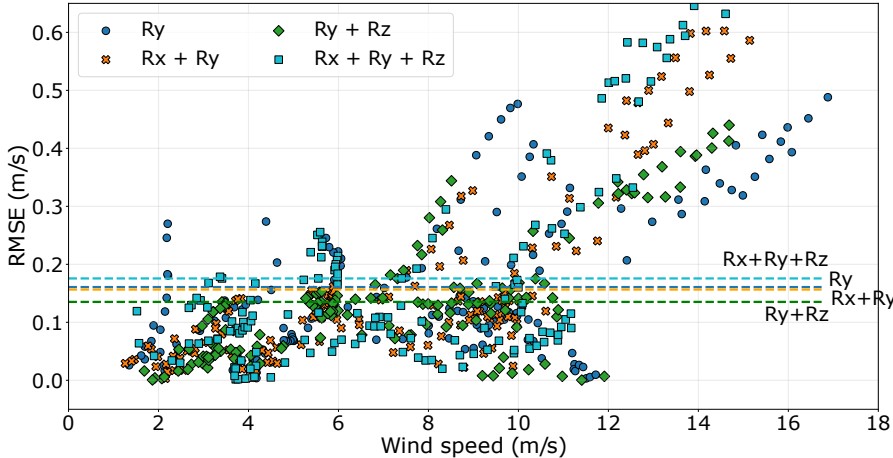

**Figure 11.** Scatter plots depicting the RMSE of the LOS velocity fluctuations measured by beam 1 at all measurement heights, plotted against wind speed. These results are derived from a set of four sequences consisting of rotations around the y-axis (Ry) and three coupling motions (Rx/Ry, Ry/Rz, Rx/Ry/Rz), all having the same movement amplitude (5 deg.) and period (6 s). Colored horizontal dashed lines represent the mean of each distribution. These lines are accompanied by labels to facilitate the interpretation of the figure.





**Figure 12.** Wind rose illustrating the RMSE of the LOS velocity fluctuations measured by beam 1 at all measurement heights. The four plots correspond to the four sequences involving rotations around the y-axis (Ry) and three coupling motions (Rx/Ry, Ry/Rz, Rx/Ry/Rz), all having the same movement amplitude (5 deg.) and period (6 s). The four numbers displayed indicate the orientation of the lidar's first four beams. Additionally, two arrows labeled x and y represent the orientation of the hexapod.

The results also demonstrate that the rotation around the vertical z-axis is not negligible in terms of RMSE. The inclusion of this rotation, Rz, in comparison to sequences without it, leads to higher RMSE values. Specifically, the RMSE generated by rotations around the y-axis (Ry) alone is lower than the RMSE generated by the combination of Ry and Rz. Similarly, the



RMSE generated by rotations around the x-axis and y-axis (Rx + Ry) is lower than the RMSE generated by the combination
of Rx, Ry, and Rz.

Similar to the single rotation Ry around the y-axis, the coupled rotation Rx/Rz around the x and z-axes results in high RMSE
when the wind is perpendicular to the y-axis, while it yields low RMSE when the wind aligns with this axis (Fig. 12). However,
the rotations Rx/Ry and Rx/Ry/Rz do not exhibit a consistent pattern. It is challenging to identify a clear influence of wind
direction on these motions. Indeed, when the wind aligns with either the x-axis or the y-axis, it can generate both high and low
RMSE values.

## 4   Discussion

An experimental campaign was conducted to evaluate the influence of motion on LOS velocity fluctuations derived from lidar
profiler measurements. The tests were conducted onshore, utilizing a WindCube v2.1 lidar mounted on a hexapod designed to
replicate the motion of a buoy. As a reference, another fixed lidar of the same type was installed nearby.

A comprehensive wind dataset spanning 45 hours, comprising 15 cycles of 3 hours each, was recorded during October
and November 2022. For this study, the focus was on regular motions involving rotation around a single axis, as well as
combinations of motions around multiple axes of rotation. This analysis pertains to half of the collected dataset (the first 9
sequences, i.e. 1.5 hours, of each cycle).

In the case of rotation around a single axis, the primary factor influencing the results was found to be the alignment between
the wind direction and the axis of rotation. The highest RMSE values were associated with winds propagating perpendicular
to the axis of rotation, causing the mobile lidar to lean in the direction of the wind. This type of motion is referred to as pitch
motion in the study by Kelberlau and Mann (2022). On the other hand, the lowest RMSE values were observed when the wind
propagated along the axis of rotation, resulting in the mobile lidar leaning perpendicular to the wind direction, known as roll
motion. These findings, which primarily focus on fluctuations and thus turbulence, are consistent with the conclusions drawn
in the study by Kelberlau and Mann (2022), which investigated the impact of motion on lidar-derived mean wind speed.

The impact of yaw motion, i.e., the rotation of the mobile lidar around the vertical z-axis, has also been evaluated by
comparing scenarios involving this rotation with equivalent scenarios that do not include it. The study has showcased that
this motion introduces short-term velocity fluctuations. Despite the generally low restoring forces associated with yaw motion,
leading to correspondingly low motion frequencies, these fluctuations notably contribute to an increase in RMSE.

Moreover, and not only considering yaw, it has been observed that the addition of motion around extra axis of rotation
increases the RMSE. It is an important finding since real FLS are often submitted to rotation around multiple axes but also
translations. Translations have not been investigated in this study due to the limitations of the hexapod. Whenever the FLS
exerts pitch and roll motions, translational motions also occur, mostly in surge and sway directions, i.e., translational motion
in the wind direction and perpendicular to it, respectively. It is anticipated that the influence of translational motion will vary
depending on its oscillation frequency relative to the lidar's sampling rate and its peak velocity relative to the wind speed.
Kelberlau and Mann (2022) demonstrated that periodic heave (vertical motion) has the most significant impact on increasing





mean wind speed estimates when it synchronizes with the lidar prism frequency. Additionally, the effect of sway motion on mean wind speed has demonstrated to be most pronounced when its oscillation frequency is low. However, further confirmation is required concerning the impact on velocity fluctuations.

Results presented in this paper have also demonstrated that the magnitude of the wind speed is one of the main driver of the RMSE. Higher wind speed leads to higher RMSE. Usually, mean wind velocities increase with elevation due to decreasing influence of surface roughness (Elkinton et al., 2006). Since the RMSE is governed by the magnitude of wind speed, it could explain why RMSE was also found to increase with increasing elevation. Such results could have meant that the wind shear is also an important driver of the RMSE of the LOS velocity fluctuations. However, no clear evidence of the role of the wind

shear could have been demonstrated.

The impact of motion period was also investigated, with periods set at 4, 6, and 8 seconds, corresponding to frequencies of 0.25, 0.167, and 0.125 Hz, respectively. These frequencies are lower than the sampling rate of the LOS velocity measurement, i.e. 1 Hz, but not negligible so as lidar measurements are performed within probes that can be considered frozen. It has been shown that these frequencies affect the LOS velocity spectra within the expected inertial sub-range, theoretically following the

classic -5/3 slope as a manifestation of the energy cascade.

However, in practical scenarios, this slope deviates from the expected -5/3 slope due to contamination of the frequency range by the probe-volume averaging effect. Therefore, in order to develop a reliable motion-compensation algorithm based on spectral analysis, aiming to achieve lidar-derived TI similar to anemometer-derived TI, a dual task must be addressed: firstly, identifying and filtering out the spectral energy associated with the motion, and secondly, tackling the loss of information

caused by the probe-volume averaging effect. These challenges are crucial in ensuring the accuracy and comparability of lidar-based measurements with traditional anemometer-based measurements.

The experiment conducted in our study involved defining scenarios with different FLS motion amplitudes: low amplitude (5 deg.) supposed to be representative of small buoy, and high amplitude (15 deg.) supposed to be representative of larger or very larger floaters. The scenarios associated with the larger floater clearly demonstrated RMSE values several times higher than

those associated with the scenarios of the small buoy.

Based on these findings, several potential recommendations can be made to improve the accuracy and reliability of lidar measurements in offshore environments. Firstly, it appears that using a small buoy as the mounting platform for the lidar may yield more favorable results in terms of reduced motion-induced turbulent fluctuations. Therefore, when possible, utilizing a small buoy or similar stable platform could be considered for future offshore lidar deployments.

Secondly, the large amplitude of the floater's motion seems to have a significant impact on the accuracy of lidar measurements. Reducing the amplitude of motion or employing stabilization mechanisms for the floater may help reduce the uncertainty on the turbulence measurement, leading to more precise and reliable data.

Furthermore, additional research and development could be focused on motion-compensation algorithms specifically tailored to address the challenges posed by large floating platforms. By accounting for the motion-induced effects in data processing, the

accuracy of lidar-derived measurements could be significantly improved, enhancing the applicability of floating lidar systems in offshore wind resource assessments and related applications.



Lastly, conducting further experiments with varying motion amplitudes and different stabilization techniques could provide valuable insights into optimizing floating lidar deployments and achieving more accurate turbulence estimates. Collaborative efforts between researchers, engineers, and industry stakeholders would be instrumental in advancing the understanding and
application of floating lidar technology.

## 5  Conclusions

The motivation for the results presented in this paper arose from a noticeable absence of experimental testing in the literature, specifically addressing the evaluation of motion-induced effects on turbulent velocity fluctuations measured by a FLS. Understanding and quantifying the impact of motion on turbulent velocity fluctuations, as measured by FLS, are of paramount
importance. The conducted experiment has facilitated a comprehensive analysis of motion-induced effects, paving the way for improved calibration and interpretation of FLS-derived wind turbulence data. However, to establish the validity of the findings from the present onshore experimental campaign, it is essential, as a further step, to conduct validation studies in real-world offshore environments or perform similar analysis on historical dataset.

The dataset presented in this paper will serve as a foundation for the formulation of a preliminary motion-compensation
algorithm, which will be addressed in further research. This algorithm will then be tested on data collected during real-world deployments of FLS at sea, alongside traditional anemometers, which are still considered the reference for turbulence measurement. Through such comparative analysis, the accuracy of the proposed motion-compensation algorithm and its effectiveness in estimating turbulence can be validated. This process will be integral to refining and optimizing the algorithm's performance to ensure its applicability and reliability in practical offshore conditions.

Furthermore, conducting a comprehensive investigation into novel stabilization mechanisms and motion-damping techniques is highly recommended as a crucial next step to mitigate motion-induced turbulence error estimates. Such an investigation is essential to significantly enhance the precision of floating lidar measurements. By integrating the developed motion-compensation algorithm with these innovative stabilization mechanisms, specifically designed for motion reduction, there is promising potential to greatly improve the accuracy of FLS-derived turbulence wind data.

The incorporation of such refined data, potentially equivalent to anemometer-derived turbulence measurements, during the design phase of offshore wind farms, will be instrumental in optimizing layout and turbine positioning. The availability of accurate and reliable turbulence wind data is vital for maximizing energy production and minimizing uncertainties throughout the planning and construction stages of offshore wind projects. Embracing these advancements will not only enhance the overall efficiency and viability of offshore wind energy projects but also contribute to the accelerated growth of offshore wind
energy on a global scale, with particular significance in countries like France, known for its substantial offshore wind potential (Marcille et al., 2023).



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

**Appendix**

| Axe | Course | Speed | Acceleration |
|-----|--------|-------|--------------|
| $T_x$ | ± 460 mm | ± 1 m/s | 10 m/s$^2$ |
| $T_y$ | ± 460 mm | ± 1 m/s | 10 m/s$^2$ |
| $T_z$ | ± 460 mm | ± 0.65 m/s | 8 m/s$^2$ |
| $R_x$ | ± 30° | ± 50°/s | 500°/s$^2$ |
| $R_y$ | ± 30° | ± 50°/s | 500°/s$^2$ |
| $R_z$ | ± 40° | ± 70°/s | 700°/s$^2$ |

**Table A1.** Hexapod's movement amplitude and dynamical performances. $T_j$ represents a translation along the j direction, $R_k$ represents a rotation around the k axis.

**Author contributions**

NT performed the analysis on the supervision of MT. MT identified the problematic and drafted the paper. GD, MLB, CM and MT designed the experiment. MLB, CM, CB and FG reviewed the paper.

**Code and data availability**

The data is owned by a public-private consortium with proprietary rights and confidentiality obligations, precluding its sharing
alongside this paper.





## Competing interest

The contact author has declared that none of the authors has any competing interests.

## Acknowlegments

We would like to acknowledge the team at Vaisala, including Mathias Régnier, Loïc Mahe, Frédéric Delbos, and Hugues
Portevin, for their invaluable support in providing and configuring two custom lidars capable of sampling at a four-times faster
rate than the commercial version. We also extend our gratitude for lending us the fixed lidar used in the present experiment.

## Financial support

This work was made possible through the support of France Energies Marines and the French government, managed by the
Agence Nationale de la Recherche under the Investissements d'Avenir program, with the reference ANR-10-IEED-0006-34.
This work was carried out in the framework of the POWSEIDOM project.