# Peer review of "Experimental evaluation of the motion-induced effects on turbulent fluctuations measurement on floating lidar systems"

_Wind Energy Science, 2023_

## Author Comment (AC1)

*General comments:*

The submitted manuscript "Experimental evaluation of the motion-induced effects on turbulent fluctuations measurement on floating lidar systems" by Thebault et al. describes a study where two WindCube profiling wind lidar units were located next to each other. One of the two lidars was mounted on a hexapod platform so that it was subject to controlled motion while wind measurements were taken. The radial wind speed data from both units were compared and differences are attributed to the effects of motion.

The study provides some novelty in so far that controlled-motion experiments of wind lidar are not often described in literature. And the study design is successful in achieving statistically relevant results because each motion sequence is performed in a variety of wind conditions, so that the effects of motion cases can be analyzed independently from the prevailing wind conditions.

Unfortunately, the depth of analysis in the current version of the manuscript is too low. Many effects are described without substantial interpretation and several aspects lack a conclusion supported by the findings of the study. The presented experiment is valuable for the FLS sector, but its potential is not well used by Thebault et al. I am missing any kind of simulation, model or at least some theoretical assumptions that predict the results. Such predictions could be validated by the experiment and in a future study be used to extend the findings to more motion cases. I will detail this in the specific comments.

I recommend reconsidering the manuscript for publication in WES only after a complete revision. A revised manuscript must demonstrate a deeper understanding of the results based on theoretical considerations of how a profiling wind lidar samples radial velocities under the influence of motion. Without this, the scientific quality is insufficient for publication in WES.

We would like to express our appreciation for your thoughtful and constructive review of our manuscript. We have carefully considered your comments and suggestions and made significant revisions to address the concerns raised.

In response to your request for a deeper theoretical understanding of our experimental results, we have added a new subsection (Section 2.4) to the manuscript. In this section, we briefly describe several factors that influence turbulence measurements obtained from FLS and discuss their expected impacts on FLS-derived turbulence measurements. While we have refrained from including a full simulation or model, we believe that this theoretical aspect will provide a clearer framework for interpreting our experimental findings. We have taken your advice and tried to offer a more robust foundation for understanding the outcomes of our controlled-motion wind lidar experiments.

Additionally, we have reworked the manuscript to provide a more in-depth analysis of our results. We have incorporated detailed interpretations of the observed effects, ensuring that readers gain a comprehensive understanding of our findings. By doing so, we hope to enhance the scientific quality and readability of the manuscript.

We acknowledge the importance of simulations and modeling, but for the current study, we have chosen to focus solely on the experimental aspects of controlled-motion wind lidar. Our decision aligns with our intention to provide practical insights that are directly applicable to the field.

We think that our paper should exclusively focus on presenting experimental results. The driving force behind this research was to address the noticeable gap in the existing literature, where experimental studies in this domain have been relatively scarce. Notably, numerous prior works have concentrated on numerical assessments of wind lidars (both FLS and nacelle-based) in their capacity to measure mean wind properties (Kelberlau and Mann, 2022; Grafe et al., 2023). Our findings are generally consistent with the outcomes of these studies (even if we are not addressing mean wind properties), except for instances where the influence of wind shear on our results remains less apparent.

In the updated manuscript, we have made a concerted effort to underscore the significance of anchoring our study solely in experimental results. We have reworked the introduction to provide a comprehensive context, shedding light on the shortage of experimental research in the literature and elucidating the critical importance of addressing this gap. In the conclusion section, we have highlighted the need for further research, specifically suggesting the incorporation of numerical analyses to validate our findings.

Furthermore, our motivation for crafting this paper extends to the presentation of a valuable dataset that we believe holds substantial promise for the wind energy community. While we acknowledge that this dataset cannot be shared publicly, we are open to sharing it within the framework of collaborative projects such as Interreg or Horizon Europe. Our vision is to establish a consortium comprising experts in experimental methods, alongside academic partners who specialize in numerical modeling. This collaborative approach will enable us to progress to the next phase of exploration: simulation and modeling, aiming to corroborate our results.

We hope that the revisions we have made address your concerns and improve the overall quality of our manuscript. We are confident that these updates provide a stronger foundation for understanding our results and their implications in the context of profiling wind lidar under the influence of motion.

***Specific comments:***

**Abstract:**

The abstract should clarify that the study is about "wind lidar measurements" to make it easier for the reader to find out if the study is relevant to them. Especially because the title of the manuscript does not give this information.

We incorporated the term "wind" before "lidar measurements". Page 1, line 2.

**Introduction:**

The introduction should differentiate better between FLS measurements of mean wind statistics and turbulence characteristics (e.g., l. 22) and research findings should be reported more precisely (e.g., l. 31, cross-contamination can lead to both over- or underestimation of turbulence).

We added the following text to fulfill your request: "Moreover, FLS serve as invaluable tools for characterizing turbulence, shedding light on complex wind flow patterns and turbulence behavior in offshore regions. This differentiation between mean wind statistics and turbulence characteristics underscores the versatility of FLS as instruments that can cater to a wide range of wind energy and meteorological research requirements in offshore and deep-water wind projects." Page 2, lines 27-30.

Also, we added the following text to report more precisely the cross-contamination effect and the probe-volume averaging effect: "Turbulence measurements obtained from lidar profilers are subject

to two main systematic errors induced by the intra-beam filtering effect, i.e., the averaging effect of the probe volume and the inter-beam filtering effect, also known as the cross-contamination effect (e.g., Peña and Mann, 2019; Kelberlau and Mann, 2020). The intra-beam filtering effect is a consequence of the probe length, effectively acting as a low-pass filter. This phenomenon stems from the filtering out of eddies that fall beneath the size threshold set by the probe length, generating underestimation of turbulence metrics. The inter-beam filtering can lead to either under- or overestimation of turbulence metrics. This discrepancy arises from the modulation of energy associated with eddies characterized by specific wavenumbers.". Page 2, lines 34-40.

Controversial or unclear statements that do not built up on a reference should be avoided in the introduction (e.g., l. 38, "high-frequency (in the range of the wave frequencies)", ll. 44-45 "...turbulence intensity is commonly assessed by calculating the variance of the three (!) velocity components...".)

We have made substantial revisions to the introduction. We have eliminated the mentioned sentences.

The authors should describe the most important findings of the cited literature and how they influence their own work instead of simply listing references (l. 40, l.43).

This part mentioned works relative to mention-compensation algorithms. We decided to remove it since the present paper does not address this topic.

A large fraction of the introduction describes studies that have a focus on mean wind statistics (ll. 62-77) without elaborating how these studies relate to the present work which presents no mean data.

We added the following text: "These studies provide valuable insights into the impact of motion-induced effects on wind measurements. While their focus primarily rests on mean wind statistics, they offer a foundational understanding of the factors that affect the accuracy and reliability of lidar-based measurements in various motion scenarios. These prior investigations serve as a crucial backdrop for comprehending the complexities and challenges associated with mitigating motion-related errors in lidar-based wind measurements." Page 3, lines 58-62.

The introduction is missing a paragraph that guides the reader through the paper while demonstrating its structure of sections.

We added a subsection "Structure of the work" at the end of the introduction. Page 3, lines 79-87.

**Data and method:**

l. 102: More detail must be given on the sampling pattern of the "prototype configuration". For how long does each beam sample in each direction within its 1Hz scanning cycle? What is the zenith angle of the beams? What are the range gates?

We added the following text: "The LOS velocity data from a standard commercial WindCube v2.1 lidar typically operates at a sampling rate of 0.25 Hz. However, for this experiment, a prototype configuration of the lidar, featuring a fourfold increase in sampling rate to 1 Hz, was employed. This improvement was achieved by significantly reducing the accumulation time for data collection from each beam by a 70%, in conjunction with a corresponding 70% reduction in the number of transmitted pulses. The elevation angle and probe length of the prototype configuration match the commercial configuration, which is 28° and approximately 23 m, respectively. Before the deployment detailed in this paper, both lidars underwent an independent performance verification conducted at the DNV Remote Sensing test site in Janneby, Germany, involving comparison against a meteorological mast."

The decision to enhance the sampling rate aimed to elevate the accuracy of turbulent fluctuations measurement, expecting to capture smaller eddies and their turbulent energy. A more comprehensive exploration of this research is anticipated to be submitted for publication in the near future.". Page 4, lines 102-111.

l. 111: More information needs to be given on the scenario with "coupled motion". Are pitch and roll motion in phase or with a phase shift or are the motions performed consecutively? This is crucial for the interpretation of the results.

Thank you for this valuable point. We added the phase for the coupled motions in Table 2. The phase of Ry was fixed at 0, Rx was set to $\pi$; whereas Rz, was set to $\pi/2$. Page 9, line 155.

l. 117: Is the availability based on 10-minute averages? It is hard to believe that none of the LOS measurements were invalid. What CNR and packet count thresholds have been used to reach 100% availability at all heights? It would have been interesting to investigate the results in challenging atmospheric conditions with bad CNR values.

We conducted a thorough verification process to ensure that both lidars consistently achieved 100% availability throughout the entire 45-hour deployment period. Notably, the custom lidar system, tailored by Vaisala, exhibited a CNR threshold of -21.5 dB, which is higher than the standard commercial configuration whose threshold is set at -23 dB. We have added a paragraph addressing this point. Page 7, lines 127-131.

We achieved a 100% availability by clearing the windows of both lidars before performing each 3-hour measurement cycle. Additionally, we refrained from collecting measurements during rainy periods, as the hexapod is sensitive to humidity, a factor known to significantly reduce data availability. Notably, the mobile lidar used in this experiment is currently deployed on Le Planier Island in the Mediterranean Sea, where data availability has experienced a noticeable decline.

Table 2: "Corresponding Scenario" descriptions are inconsistent (e.g., S2 T=4s "Typical large buoy" but also S4 T=6s "Large (or spar) buoy" or S3 and S5 have different periods but the same description and what does "weak response above" mean?). Typical commercial FLS have tilt response periods of around 3s. Periods above 4 seconds are rather found for larger platforms.

Thank you for your remark. We removed the column "Correspond scenario". It was confusing for the reader. We also removed the part of the discussion associated with the different scenarios.

l. 127: Commercial FLS usually use simple single point moorings. Thus, no valid evidence is provided for the important statement that the motion characteristics of the MONABIOP buoy are representative for the motion of FLS.

You are right. We modified the text which is now: "While it's important to note that commercial FLS often use single-point moorings, MONABIOP employs a distinct mooring system with three semi-taut lines to restrict its motion. This design is primarily intended for testing and validating a mooring system that utilizes nylon ropes. It's worth acknowledging that while MONABIOP's motion dynamics may not precisely replicate all aspects of commercial FLS systems, it provides valuable insights into the behavior of buoy-like structures in dynamic marine environments.". Page 8, lines 140-144.

l. 135: Which "specific conditions"? This description is too general and does not add any value.

We removed the term "specific conditions" and wrote instead this sentence: "Amplitudes of 5 deg. and 15 deg. were selected to represent medium and high tilt motions, respectively as proposed in Kelberlau and Mann (2022)". Page 8, lines 149-150.

l. 144: RMSE is used as the key parameter in this study. Its definition must be given by a well-described formula.

We added a formula (Eq. 1) to define RMSE. Page 9, line 164.

l. 145: Which method was used for getting the "mean-detrended signal"?

We added the following text: "Turbulent velocity fluctuations were assessed by computing the standard deviation, denoted as σ, from the mean-detrended signal derived from 10-minute ensembles of the LOS velocities. This process involved the removal of the mean value (i.e., the constant component), effectively centering the data around zero through subtraction of the mean from the original signal.". Page 9, lines 157-159.

**Results:**

l. 155: It would be much easier for the reader to see how well the low frequency component (!) of the time series align if they were given in the same plot instead of in two separate plots.

Thank you for your remark. We modified the figure (now Fig. 5) according to your suggestion.

Fig. 4: Instead of presenting and comparing data from two separate plots the data from both lidars should be presented in one plot (maybe just one or two beams). A second plot could then be used for a zoomed-in section that also shows the motion period for comparison.

Please, see the comment just above.

l. 163: The authors note that the standard deviation of LOS velocity fluctuations is 70% higher for the mobile lidar than for the fixed lidar. But as a reader I am missing an interpretation of this value. Is this what is expected from the beam rotation with the chosen amplitudes? In a completely homogeneous wind field of a fixed wind speed, the effect of pitch rotation on a single beam can be estimated. It would be of outmost interest to compare the experimental fluctuations with these theoretical fluctuations.

We did not anticipate a precise value of 70%. In the revised manuscript, this value is utilized to compare the 0.8% difference in LOS velocity fluctuations between measurements obtained from the mobile lidar and the fixed lidar during the period when both lidars were stationary.

While your suggestion to compare experimental fluctuations with theoretical ones is pertinent, we have decided to concentrate exclusively on the experimental aspects in this paper. We intend to reserve such an analysis for future research, which will be conducted by our team or in collaboration with research institutes and academics.

l. 167: "nearly three times higher". I think in l. 174, the authors write about the same numbers that they are "more than four times higher".

It was correct but probably unclear. First, we are talking about the amplitude and second, about the RMSE. We rewrote this part.

l. 170: The analysis of the impact of motion amplitude is missing a comparison of RMSE in the absence of motion. The beams of both lidar units will not measure identical LOS speeds even when they are both standing still (not synchronized, not the same air volumes, not the same angles, random measurement error...). This analysis could also be added to a previous section.

You are right. The second reviewer also required the addition of such analysis. We have added the following text in section 3.1 – Preliminary results: "The analysis begins by examining the turbulent fluctuations measured by both lidars, without motions. Throughout the entire measurement campaign, these lidars operated independently without synchronization and were positioned 10 meters apart, leading to the measurement of distinct air volumes. Such disparities in measurement can potentially introduce gaps in the estimation of the standard deviation of LOS velocity. The average σ, associated with beam 1, as measured by the fixed lidar, was determined to be 0.719 m/s. In contrast, the average σ obtained from the mobile lidar measurements was slightly higher, specifically by 0.8%, resulting in a value of 0.713 m/s. A graphical comparison of σ derived from both lidars can be observed in Fig. 4a." Page 10, Lines 202-210.

We have also included a new figure (Fig. 4a).

l. 174: Without a simulation model, the interpretation of the results is superficial. The authors state a "linear increase" of RMSE with wind speed. A FLS model will probably show that this statement is only true in the absence of translatory motion of the lidar telescope which is introduced here by its rigid body rotation around a non-zero lever. So, the linear relationship between RMSE and wind speed is only an approximation that should be assessed critically.

You are correct. This observation should be validated through a thorough numerical analysis. We have deferred this task for future research and, as a result, have removed the mentioned sentence.

l. 185: What could be a reason for the higher spectral energy at low frequencies measured by the fixed lidar? This is a crucial finding that must be investigated further because it has an impact on the RMSE values. It is insufficient to conclude with "distinct characteristics in the spectral energy profiles".

Your point is quite intriguing. In response, we have included spectra corresponding to a 5-degree period for comparison with those of a 15-degree period (refer to Fig. 7 in the updated version of the manuscript). You will notice that, for a 5-degree amplitude, the spectra obtained from both the fixed and mobile lidar are nearly identical, in contrast to the spectra associated with a 15-degree amplitude.

We added this text:

"The spectra obtained from the mobile lidar clearly exhibit a spike in energy corresponding to the rotation frequency. The height of this spike remains consistent for each motion period and is lower for the lowest amplitude. For both amplitudes, the spectral energy measured by the mobile lidar surpasses that of the fixed lidar for the higher frequencies. Moreover, this difference in spectral energy becomes more pronounced for the lowest motion period. Conversely, at lower frequencies, the spectral energy associated with a 15-deg. amplitude, derived from measurements of the fixed lidar, consistently surpasses that of the mobile lidar. In the case of a 5-deg amplitude, the spectral energy derived from measurements of both fixed and mobile lidars shows overlap." Page 12, lines 234-240.

And a possible explanation in the discussion:

"The study highlights that turbulence measurements obtained from FLS are more sensitive to changes in orientation (amplitude of motion) than to motion periods. This finding underscores the significance

of changes in measurement geometry due to platform orientation. The analysis revealed a strong correlation between high RMSE and high amplitude. Additionally, it was observed that amplitude significantly influences the measurement of spectral energy, particularly in the low-frequency domain, associated with high turbulence length scales. When the lidar system tilts, it effectively acquires data from diverse air masses and turbulence conditions, resulting in fluctuations in turbulence measurements". Pages 19-20, lines 334-339.

Fig. 8: The spectra should contain a vertical line at the frequencies that correspond to the motion periods. Otherwise, it is difficult to analyse the spectral peaks.

We modified the figure according to your suggestion. Fig. 7 in the updated version of the manuscript.

3.5: The authors present some good hypothesis for why measurements at high elevations show higher RMSE values. Also here, a model framework and an in-depth analysis of example time series would help to quantify the influence of the different sources of added RMSE. Without it the findings are inconclusive.

All the findings presented in this paper remain inconclusive as they require further validation through numerical analysis, which on its own also presents limitations. We have discussed this aspect in the conclusion section:

"However, to establish the validity of the findings from the present experimental campaign, it is essential, as a further step, to conduct numerical modeling to compare with experimental results. This represents a promising avenue for forthcoming research. Numerical models can serve as valuable tools for validating and complementing experimental findings. They can help provide a deeper understanding of the underlying physical mechanisms and allow for simulations under controlled conditions, enabling the exploration of a broader range of motion scenarios. Future studies should consider integrating numerical models to enhance the robustness of the conclusions." Page 20, lines 359-364.

3.6: This section lacks a conclusion. How does wind shear influence the RMSE of the mobile lidar?

It appears that wind shear has no discernible effect on the RMSE of the mobile lidar. We did not find any conclusive evidence to the contrary. This lack of effect is not consistent with the observations made by Kelberlau and Mann (2022), where they highlighted the clear impact of wind shear on the measurement of mean wind properties using FLS.

Throughout the abstract and discussion sections, we have emphasized the need for additional analysis, potentially involving numerical modeling or further measurement campaigns, to draw definitive conclusions regarding the influence of wind shear.

l. 223: Why does this observation hold only in the range from 12 to 14 m/s? The mean values of all wind speeds show a different order (dashed horizontal lines in Fig. 11, Ry > Ry+Rz). Without further explanation, it is not convincing that "the rotation around the vertical z-axis is not negligible in terms of RMSE".

We completely rewrote this subsection. Pages 16-18, Lines 280-296.

**Discussion:**

The weaknesses of the manuscript mentioned above result in a discussion that is superficial for the

most part (e.g., l. 265 "wind speed is one of the main drivers[s] of the RMSE". This is obvious from theoretical considerations even without any experiment, l. 269 "no clear evidence of the role of wind shear could have been demonstrated."). Other statements are not covered by the evidence presented in the previous sections (e.g., ll. 257-264, findings regarding the frequency dependency of effects of translational motion are only valid for reconstructed wind vectors, not for RMSE of LOS beams.).

We thoroughly revised the discussion, delving deeper into our findings regarding how a lidar measures LOS velocities under motion. Pages 18-20, lines 297-352.

Other sentences are confusing or wrong (e.g., ll. 282-285: small buoys, small amplitudes and large buoys, large amplitudes). The recommendation to use small buoys as FLS platform to reduce motion induced errors is likewise misleading.

We removed this part.

**Conclusions:**

l. 325: It remains unclear why the conclusions of the study are more significant for France than for other countries with similar offshore wind potential.

We rewrote the conclusion and removed this last sentence.

*Technical corrections:*

The manuscript contains several minor errors and has room for stylistic improvement. A revised version should be proofread carefully before submission.

We have addressed the minor errors and refined the manuscript's style. We have also conducted a thorough manual review of the entire manuscript.

**Bibliography**

Kelberlau, F., & Mann, J. (2022). Quantification of motion-induced measurement error on floating lidar systems. *Atmospheric Measurement Techniques*, *15*(18), 5323-5341.

Gräfe, M., Pettas, V., Gottschall, J., & Cheng, P. W. (2023). Quantification and correction of motion influence for nacelle-based lidar systems on floating wind turbines. *Wind Energy Science*, *8*(6), 925-946.

---

## Author Comment (AC2)

Dear authors,

Thanks for the manuscript. I think in general the manuscript is well written, coherent and the contents relate to a very interesting topic, which is that of the floating lidar turbulence measurements. However, although the manuscript presents a quite interesting dataset that can be used to analyze the impact of motion-induced effects on lidar turbulence, I think that the manuscript reads more as a technical report than a journal paper. Below I will provide some general and specific comments with respect to this and different aspects of the study.

General comments

1. As I mentioned, right now we are reading an interesting technical report but not a research paper. The reader is not gaining anything new from the paper as the data and some analyses are presented without further investigation. The authors mentioned that they are going to propose a motion-compensation algorithm based on this dataset. I think that that is what this report needs to have potential for a paper, so I encourage the authors to start the paper by the planned algorithm to compensate for motion and investigate its goodness using this dataset.

We appreciate your review and your constructive feedback regarding our manuscript. In response to your insights, we have undertaken substantial revisions to bring the manuscript more in line with the conventions of a scientific paper, moving away from its earlier semblance to a technical report.

It is crucial to clarify our current paper's primary focus. We no longer address the topic of motion-compensation algorithms in the updated version of the manuscript. The initial version of the paper may have given a misleading impression in this regard.

Presently, we are not prepared to introduce an algorithm that outperforms the established methods found in the existing literature. Our central objective with this paper is to introduce and comprehensively present the dataset we have collected. Our focus is twofold: first, to identify the main sources of error in turbulence measurements using a FLS; and second, to address a conspicuous gap in the current literature. This gap largely stems from the scarcity of experimental studies in this domain, which have predominantly focused on numerical analyses, particularly in relation to mean wind parameters. Our next step involves validating our results through numerical analyses.

We envision this dataset as a valuable resource, one that can be readily shared within a consortium of upcoming (European) research projects, particularly those that are dedicated to the development of motion compensation algorithms. It is our hope that this dataset will play a pivotal role in advancing research in this field.

2. Line 31: it is nice you are aware that there are two main errors for lidar turbulence measures (most people do not know this) but it is also strange that you think that the cross-contamination effect always results in an overestimation. This is not the case always. If by some reason your compensation algorithm always assumes an overestimation due to cross-contamination, you need to review it deeply.

You are right. We have modified the text: "The inter-beam filtering can lead to either underestimation or overestimation of turbulence metrics. This discrepancy arises from the modulation of energy associated with eddies characterized by specific wavenumbers.". Page 2, lines 38-40.

3. Table 3: I am not sure of the value of this table. The "degree" of deviation of the mobile lidar turbulence compared to the fixed lidar turbulence should be both turbulence and scanning-configuration dependent. Here you seem to average across all cycles which I guess have different turbulence characteristics, so you are kind of averaging apples with oranges.

Indeed, we averaged across all cycles, complicating the interpretation of the results. Consequently, we opted to exclude this table as it did not contribute meaningful value to the paper.

4. One important question: did you by chance make a cycle without motion at all? That would be interesting to have as part of the analysis to know whether there is an inherent bias between the units.

Certainly, we have recorded such a cycle, capturing a 2-hour dataset during which both lidars remained stationary. We have incorporated a corresponding figure (Fig. 4a) illustrating the scatterplot of the standard deviation of LOS velocity from both lidars. This plot serves as a comparison to the scatterplot obtained by the fixed lidar and the ostensibly 'mobile' lidar in motion (Fig. 4b). Page 10, lines 202-210.

5. Section 3.3/Figure 8: there should not be that much difference between the spectra of the mobile and the fixed lidar (particularly at the large scales) apart from the area around the peak at the specific period. Why is it different (see my previous comment)? Maybe some error bands could show that these differences are not significant as they seem to be?

We have addressed this crucial point also highlighted by the second reviewer. We have conducted an in-depth investigation into the spectra and the impact of amplitude on the level of spectral energy. In the previous version of the manuscript, we exclusively presented spectra associated with a 15-deg. amplitude. Our findings revealed that, at lower frequencies, the spectral energy for LOS velocity derived from the fixed lidar surpassed that from the mobile lidar.

The tilting of the lidar system introduces data acquisition from diverse air masses and turbulence conditions, leading to fluctuations in turbulence measurements. This phenomenon may account for the observed disparity in spectral energy between measurements from both lidars. In the updated manuscript, we have enhanced Fig. 7 with additional panels (a, b, c), illustrating spectra associated with a 5-degree amplitude. Notably, these spectra demonstrate an overlap in spectral energy derived from measurements of both lidars at this amplitude, within the low-frequency domain (large scales).

We added this text:

"The spectra obtained from the mobile lidar clearly exhibit a spike in energy corresponding to the rotation frequency. The height of this spike remains consistent for each motion period and is lower for the lowest amplitude. For both amplitudes, the spectral energy measured by the mobile lidar surpasses that of the fixed lidar for the higher frequencies. Moreover, this difference in spectral energy becomes more pronounced for the lowest motion period. Conversely, at lower frequencies, the spectral energy associated with a 15-deg. amplitude, derived from measurements of the fixed

lidar, consistently surpasses that of the mobile lidar. In the case of a 5-deg amplitude, the spectral energy derived from measurements of both fixed and mobile lidars shows overlap." Page 12, lines 234-240.

And a possible explanation in the discussion:

"The study highlights that turbulence measurements obtained from FLS are more sensitive to changes in orientation (amplitude of motion) than to motion periods. This finding underscores the significance of changes in measurement geometry due to platform orientation. The analysis revealed a strong correlation between high RMSE and high amplitude. Additionally, it was observed that amplitude significantly influences the measurement of spectral energy, particularly in the low-frequency domain, associated with high turbulence length scales. When the lidar system tilts, it effectively acquires data from diverse air masses and turbulence conditions, resulting in fluctuations in turbulence measurements". Pages 19-20, lines 334-339.

6. Lines 309—314: these lines cannot be part of the conclusions. You have not described the motion-compensation algorithm and you are here giving us hints of what it can do. As mentioned in my first comment, I suggest you start this manuscript by proposing/explaining the algorithm and then you evaluate it by comparison with this nice dataset.

We have revised the conclusion and systematically eliminated any references to the motion-compensation algorithm throughout the entire manuscript.

Specific comments

1. Line 25: Delete an extra ")".

Thank you. Removed.

2. Line 33: Delete "in most cases, … However,".

We removed this part when we rewrote the introduction.

3. Line 33: Replace "do align" by "compensate each other".

Similar to our comment just above.

4. Introduction: Peña et al. (2022) presented another way to assess the impact of motion on floating lidars that you would like to check.

Thank you for this valuable reference. We now mentioned this study in the introduction: "Pena et al. (2022) utilized simulated lidar profiler measurements within synthetic atmospheric turbulence fields to evaluate how buoy motions affect turbulence estimation. Their simulations revealed that translational motions of the buoy notably influenced the accuracy of turbulence estimates.". Page 2, lines 53-56.

5. Lines 163—166 are also "strange" ways to measure these impacts (see general comment 3).

We removed the part linked to Table 3. See our response to your point 3 in General Comment.

6. Line 175: "latter metric" … it is difficult to understand what do you mean here.

References:

Peña A., Mann J., Angelou N., and Jacobsen A. (2022) A motion-correction method for turbulence estimates from floating lidars. Remote Sensing, 14, 6065